# Accumbal Dopamine Responses Are Distinct between Female Rats with Active and Passive Coping Strategies

**DOI:** 10.3390/biom14101280

**Published:** 2024-10-10

**Authors:** Vsevolod V. Nemets, Ekaterina P. Vinogradova, Vladislav Zavialov, Vladimir P. Grinevich, Evgeny A. Budygin, Raul R. Gainetdinov

**Affiliations:** 1Institute of Translational Biomedicine, Saint Petersburg State University, 199034 Saint Petersburg, Russia; v.v.nemets@spbu.ru (V.V.N.); vladislavletsgo@gmail.com (V.Z.); 2Department of High Neuros Activity, Saint Petersburg State University, 199034 Saint Petersburg, Russia; e.vinogradova@spbu.ru; 3Department of Neurobiology, Sirius University of Science and Technology, 354340 Sirius, Russia; vlad.p.grin@gmail.com (V.P.G.); ebudygin@wakehealth.edu (E.A.B.)

**Keywords:** social defeat, female rats, dopamine release, nucleus accumbens, stress coping

## Abstract

There is a gap in existing knowledge of stress-triggered neurochemical and behavioral adaptations in females. This study was designed to explore the short-term consequences of a single social defeat (SD) on accumbal dopamine (DA) dynamics and related behaviors in female Wistar rats. During the SD procedure, rats demonstrated different stress-handling strategies, which were defined as active and passive coping. The “active” subjects expressed a significantly higher level of activity directed toward handling stress experience, while the “passive” ones showed an escalated freezing pattern. Remarkably, these opposite behavioral manifestations were negatively correlated. Twenty-four hours following the SD exposure, decreased immobility latency in the Porsolt test and cognitive augmentation in the new object recognition evaluation were evident, along with an increase in electrically evoked mesolimbic DA release in passive coping rats. Rats exhibiting an active pattern of responses showed insignificant changes in immobility and cognitive performance as well as in evoked mesolimbic DA response. Furthermore, the dynamics of the decline and recovery of DA efflux under the depletion protocol were significantly altered in the passive but not active female rats. Taken together, these data suggest that female rats with a passive coping strategy are more susceptible to developing behavioral and neurochemical alterations within 24 h after stress exposure. This observation may represent both maladaptive and protective responses of an organism on a short timescale.

## 1. Introduction

Social stress, which is an outcome of a single or multiple socially unfavorable traumatic events or “stressors”, considerably impacts our modern life on a daily basis. Stress-induced changes may have protective features but may result in a variety of dysfunctions, which could further be transformed into serious pathological conditions [1,2,3,4,5,6,7,8]. This includes but is not limited to anxiety, depression or post-traumatic stress disorder (PTSD). Human studies have shown a difference in gender susceptibility to social stressors. According to a National Comorbidity Survey, women are more likely to suffer major depression than men at 21.3% and 12.7% rate, respectively [2,9]. Likewise, PTSD is at least two times more common in women [10,11], while greater exposure to trauma cannot account for this difference [12]. Unfortunately, our knowledge of biological mechanisms that can potentially explain sex differences in stress-triggered physiological consequences is still very limited [11].

Animal models replicate differences in sex predisposition, as well as individual adaptation capability to stress in regard to behavioral consequences, metabolic changes and lifespan [13,14,15]. Thus, female rats are more susceptible than males to short-term stressors and chronic mild stressors and show a depressive-like condition [14]. However, they might be less sensitive than male rats to prolonged severe stressors [15,16]. Behavioral responses, which attempt to deal with challenges during a stressful situation, are often termed “coping strategies”. This behavior targets stressful stimuli in order to remove, avoid, minimize, tolerate or take them under control [6]. In dealing with stress, humans and animals may use a number of different coping strategies, although coping strategies can be divided into two general types, which are passive and active [6]. This distinctiveness is based on the presence or absence of attempts to act upon the stressor in active or passive coping mode, respectively.

Brain neurotransmission plays a pivotal role in the build-up and shaping of coping strategies [17], subsequent stress-induced behavioral and physiological alterations, and their regaining [6,18]. Consequently, certain neurochemical mechanisms should be responsible for the consequences of stress exposure in regard to resilience and dysfunctions. In particular, changes in mesolimbic dopamine (DA) observed in stress-exposed animals have been shown to be connected to both stress-adaptive processes and abnormal responses [19,20,21,22]. Thus, mesolimbic DA can be considered a key neurotransmitter involved in stress outcomes. Remarkably, the neuroadaptation in neurotransmission may take place following a single exposure to the stressor. For example, the induction of long-term potentiation (LTP) at GABA_A_ synapses on the ventral tegmental area (VTA) DA neurons was inhibited 24 h after acute stress [23]. Perhaps, since DA cells in the VTA are under powerful control of GABA neurons [24,25], the burst firing of DA neurons increases with acute restrain stress and perseveres for at least 24 h [26,27]. In agreement with this mechanism, there is recent evidence that DA release is enhanced in the terminal field (nucleus accumbens) in rats that experienced a single bout of social defeat. Noticeably, these studies were conducted exclusively in male rats. In fact, only negligible neurobiological research was performed on female rodents [11,28]. Therefore, there is a gap in existing knowledge of stress-triggered neurochemical and behavioral adaptations in females.

To this end, the current study was designed to explore specific responses of female rats to acute social stress (experiment 1) and follow-up behavioral and neurochemical (mesolimbic DA) consequences, which can be developed on a short-term (24 h) time scale (experiment 2). The data obtained in female rats extended our earlier findings in males and, for the first time, linked stress-triggered adaptations in mesolimbic DA with the coping strategy.

## 2. Materials and Methods

### 2.1. Animals

Female Wistar rats (300–350 g) and the same age (3 months) female tryptophan hydroxylase 2 knockout (Tph2-KO) Dark Agouti rats (300–350 g) were used in this study. The latter rats have genetically reduced levels of brain serotonin (5-HT) and thus are known to exhibit enhanced aggressiveness [29,30]. All animals were housed in plastic cages (40 × 60 × 20 cm, 5 rats per cage) and were maintained on a 12 h light/12 h dark cycle with food and water available ad libitum. All procedures involving animals were conducted in accordance with the *Guide for the Care and Use of Laboratory Animals: Eighth Edition* [31], and the animal study protocol was approved by the Institutional Animal Care and Use Committee at St Petersburg University (protocol No. 131-03-8, 25 September 2023).

### 2.2. Procedure of Social Defeat (SD)

To achieve stressful conditions, Wistar rats were exposed to a single episode of SD, lasting 20 min and consisting of three phases. The initial phase was to introduce the “intruder” Wistar rat to the home cage of the “resident” Tph2-KO rat (the aggressive opponent) for 5 min [32,33,34]. During this initial phase, the intruder was protected with the inset wire mesh cage, which allowed social interactions and species-typical threats by the female aggressive resident, thus facilitating the instigation of aggression. In the second phase, the inset cage was removed to allow direct confrontation between the rats for 10 min. Finally, the inset cage was put back to separate the rats once again for 5 min, allowing the resident to resume social threats. The non-stressed control animals underwent the same protocol, but aggressive “residents” were replaced with Wistar female rats. All encounters of the SD procedure were video recorded to ethologically analyze the behaviors of the intruder rats. These behaviors of the intruder rats were enumerated (duration time) during the direct physical contact phase (Figure 1B). All behavioral tests were conducted within the 1:00 p.m.–5:00 p.m. time frame in order to minimize the effects of the circadian rhythm [35].

Assessing behavior in the intruder rats during the SD procedure, their stress coping paradigms in particular [36], the intruders were divided into active and passive coping subjects based upon their active (consisted of engaged behavioral patterns such as exploration, attacks, defense, running, grooming and digging), or passive (lost behavioral pattern such as freezing) behavior (Figure 2A–E).

All experimental groups of rats consisted of an equal amount of diestrus and estrous/proestrus females, which were tested via cycle stage identification before SD or social interaction (in the control group). It was performed to avoid any influences of estrous stages on behavior or DA dynamics [37]. The estrous cycle in all female rats was assessed every day for 2 weeks before those rats were used in experimental procedures [38]. These assessments were performed around noon in favor of the estradiol daily maximum level and were accounted to achieve balanced experimental and control groups of animals [37]. The cycle stages in vaginal smears were detected cytologically by using Eosin-Methylene Blue [37,38,39].

### 2.3. Post-SD Behavioral Tests

After SD or social non-aggressive interactions, all animals were kept in individual cages for 24 h while they were subjected to either fast-scan voltammetry measurements or behavioral tests (Figure 1). Regarding behavior, all animals underwent a standard battery of behavioral tests in the following order: sucrose preference, then maze and novel object recognition, and finally, a forced swim Porsolt test as the most stressful for rodents [40,41]. In the current study, we used the same procedure scheme as we utilized in our previous study on male rats [42]; however, in the present study, we conducted additional behavioral tests as follows.

A sucrose preference test was performed in individual cages 24 h after the SD session or social interaction (control) to reveal whether an anhedonic condition could be triggered by a short-term exposure to the stress. The two-bottle choice design was used to ensure simultaneous access to water and a 10% sucrose solution for 18 h a day, 3 days a week, for 2 weeks [42]. Each day, at the beginning and the end of the drinking session, the bottles were weighed, and the amount of each ingested fluid was calculated. No food or water deprivation was applied before the test. The consumption of water and 10% sucrose was assessed during the dark circadian phase [43]. Sucrose preference was calculated as a ratio of the amount of 10% sucrose solution (g) to the sum of total consumed fluid (g) and was expressed in percentage [44].

The elevated plus maze test was used to explore the short-term consequences of the single SD on locomotion and anxiety behaviors. This test is based on rodents’ natural fear of open spaces (open arms) while, at the same time, their preference for closed ones (closed arms) [45,46]. Rats were acclimated to the testing room and then individually placed towards the center of the maze. Their spontaneous activity in the open and closed arms of the maze was registered for 5 min with a video-recording system (EthoVision XT 11.5, Noldus, Wageningen, The Netherlands). After each subject, the arena was cleaned with 3% hydrogen in order to eliminate olfactory cues. The duration of behavioral patterns (rearing, time spent in closed arms) was measured manually by analyzing the recorded videos.

To explore if SD causes cognitive dysfunctions, which can be observed in a depressive-like state, the novel object recognition (NOR) test was performed [47,48,49,50,51]. All procedures were carried out in individual home cages to avoid additional stress [52]. The NOR test consisted of two phases. The first one was an object-familiarizing phase when two identical objects were introduced to the rat for 8–10 min on opposite sides of the cage (object A vs. object A). The second one was an object-recognition phase when one object was replaced with a novel one, which was different in color, shape and texture (object B vs. object A). The gap between tasks was 1 h. The recognition phase lasted 3 min, during which the time spent studying each object was video recorded. Novel object recognition was identified while the animal was sniffing or otherwise exploring a novel object at a distance closer than 1 cm [53]). The discrimination index for a novel object was calculated based on the difference in time spent studying those objects using the formula (B − A)/(A + B) [53,54].

Finally, the forced swim test procedure [55,56,57] was performed to verify a stress-like phenotype in rats 24 h after SD. Briefly, female rats were allowed to acclimate to the testing room for 40 min. Then, each rat was placed in a water-filled cylindrical glass container (height was 45 cm, diameter was 28 cm) with a water temperature of 23 ± 1 °C. Rat behavior was identified for 6 min as either swimming or floating (immobility), and the latency of the first immobilization state was measured [58,59]. The floating behavior was determined as the nonappearance of any directed movements of the body or head [60]. The container was cleaned after each subject.

### 2.4. Post-SD Voltammetric Measurements of DA in the Nucleus Accumbens

Electrically evoked dopamine release was recorded by FSCV in the nucleus accumbens of anesthetized rats 24 h after the SD procedure. Rats were anesthetized by using a single intraperitoneal (i.p.) injection of urethane (1.5 g/kg) and secured in a stereotaxic frame. Holes were drilled in the scalp in order to implant electrodes into the brain. A stimulating electrode was inserted into the VTA (AP: –5.2 mm; ML: 1.0 mm; DV: 8.2–8.4 mm), a carbon fiber working microelectrode (exposed fiber length 75–100 μm; diameter 6 μm) was placed into the nucleus accumbens (AP: 1.3 mm; ML: 1.3 mm; DV: 6.6–6.8 mm) and an Ag/AgCl reference electrode into the brain tissue of the contralateral hemisphere. The electrodes were connected to the voltammetric amplifier interfaced with a computer running the specialized software. To explore the difference in frequency-dependence of dopamine release between SD and control rats, 1 s electrical stimulations (330 µA) of the VTA were made at different frequencies (5, 10, 20, 30, 50 and 60 Hz) every 10 min. The depletion protocol included three consequent stimulations (330 µA, 60Hz, 600 pulses), which were applied in 1–2 s intervals. Then, a regular stimulation (330 µA, 60 Hz, 60 pulses) was applied in 1 min (14 stimulations), 5 min (3) and 10 min (5) intervals to allow recovery of the dopamine signal. Extracellular dopamine was detected at the carbon fiber electrode every 100 ms by applying a triangular waveform (−0.4 V to +1.3 V and back to −0.4 V vs. Ag/AgCl, 400 V/s). The dopamine signal was verified by a background-subtracted cyclic voltammogram characterized by oxidation and reduction peaks occurring at +0.6 and −0.2 V, respectively [42,61,62,63].

### 2.5. Limitations

This study was not designed to directly compare the effects of social defeat exposure on behavior and neurochemical measures of female and male rats. Nevertheless, the current experiments were performed on females followed by the same social defeat paradigm, and the FSCV protocol was recently tested on males [42], which allowed us to deliberate findings from both genders to some extent together. However, we acknowledge that a direct comparison is not applicable since the experiments with male and female subjects were not executed side by side.

### 2.6. Statistical Analysis

Voltammetry-measured DA oxidation current (nA) was converted into a molar concentration of released DA (µM) or was expressed as a percentage of basal value. Statistical analysis of FSCV data was performed by applying a repeated-measures two-way ANOVA. Regarding behavioral studies, data for each behavioral element were accounted for as a percentage of total observation time. The D’Agostino-Pearson omnibus normality test was used to evaluate whether the values followed Gaussian distribution, and then, in the case of normal distribution, we used a parametric one-way ANOVA test with multiple comparisons, otherwise nonparametric Kruskal–Wallis test with multiple comparisons. All analyses were carried out using GraphPad Prism (version 6.05, San Diego, CA, USA). The data were expressed as a mean ± SEM with a criterion for significance set at *p* ≤ 0.05.

## 3. Results

### 3.1. Experiment 1: Evaluation of Behaviors during SD Session

The SD-exposed intruders were divided into two cohorts: active and passive coping groups. This separation was based on their coping strategy during the interaction with an aggressive TPH2-KO resident (see Section 2.2 for details). The coping with stress performance was also assessed in comparison with analogous behaviors observed in unstressed controls (Figure 2A–E). Thus, passively coping rats showed significantly more durable passive behavior, such as a “freezing” vs. control and active coping rats 22.4 ± 4.8% vs. 2.2 ± 0.8% and vs. 6.1± 1.7%, *p* < 0.001 and *p* < 0.01, respectively (multiple comparison Kruskal–Wallis test). Furthermore, as shown in Figure 2E, passively coping rats demonstrated sufficiently less active behavior, aka engaged behavioral patterns, such as exploration, attacking, defense, running, grooming and digging in comparison with control and active coping subjects, *p* < 0.0001 (multiple comparison one-way ANOVA). Remarkably, a significant negative correlation (Figure 2F) was found between intruders’ explorative and freezing behaviors during the SD session (r = −0.59; *p* < 0.05; two-tailed Pearson analysis).

### 3.2. Experiment 2: Behavior Alterations Observed 24 h after SD

By assessing the behavior of the female Wistar rats 24 h after being exposed to a single SD, we found significant behavioral alterations in defeated animals depending upon their different coping strategies during SD (Figure 3). We found no significant changes in sucrose preference between all tested groups (multiple comparison Kruskal–Wallis test, *p* ≥ 0.2) (Figure 3A). Surprisingly, cognitive performance, measured as a novel object discrimination ratio (Figure 3B), was significantly enhanced in passively coping rats vs. controls and actively coping subjects, *p* < 0.01 (multiple comparison Kruskal–Wallis test). There was no significant difference in the time of total objects investigation in all groups (multiple comparison Kruskal–Wallis test, *p* > 0.2). Measuring the latency of the first immobility in the Porsolt test (Figure 3C), we found a significant difference between passively coping rats vs. controls, *p* < 0.05, and a meaningful trend in passive coping rats compared to active subjects, *p* = 0.06 (multiple comparison Kruskal–Wallis test). However, the total immobility time was not different between all groups). No changes were also found in the elevated plus maze regarding the time spent in closed arms (Figure 2D). However, there was a significant tendency to increase the rearing number (Figure 3E) in both SD-exposed subgroups (multiple comparison Kruskal–Wallis test, *p* = 0.06).

### 3.3. Experiment 2 (Continued): Changes in Accumbal DA Detected 24 h after Single SD

Voltammetric measurements of DA release in the nucleus accumbens of anesthetized rats following electrical stimulation of the VTA were used to evaluate DA changes after the stressful environments of the SD procedure. In Figure 4A, representative DA effluxes observed in control and actively or passively coping with SD-evoked stress female rats are presented. Electrically evoked DA release was enhanced in all rat groups in a frequency-dependent manner (repeated measures two-way ANOVA; F (1.524, 35.04) = 95.03, *p* < 0.0001) (Figure 4B). Remarkably, passive coping animals exhibited pronounced DA responses in comparison to controls and actively coping groups (Tukey’s multiple comparisons test two-way ANOVA; *p* ≤ 0.001 and *p* < 0.01, respectively).

Following the DA depletion protocol (Figure 4C), we found prominent alterations in the DA depletion course displayed in passive coping rats vs. actively coping ones and control subjects, both significant at *p* value less than 0.0001 (Sidak’s multiple comparisons test, two-way ANOVA). The prolonged stimulation (10 s, 60 Hz) induced a significantly stronger effect on consequent evoked DA levels in passive coping subjects vs. controls (*p* ≤ 0.01, multiple comparison Kruskal–Wallis test) (Figure 4D). Similarly, DA levels in the nucleus accumbens of these rats were decreased during recovery processes (*p* < 0.05, multiple comparison Kruskal–Wallis test) (Figure 4D). In addition to neurochemical changes, we found a significant negative correlation between accumbal DA responses and the exhibition of preceding active behavior during the experience of SD (r = −0.57; *p* < 0.05; two-tailed Pearson analysis).

## 4. Discussion

The current study in female rats extends our earlier findings obtained in male rats and indicates that stress caused by a single SD is capable of generating behavioral and DA alterations in the nucleus accumbens, which are visible within 24 h. In addition, this study highlighted different stress-coping strategies of female rats during the SD procedure. Specifically, some rats expressed a higher level of activity with intense exploratory patterns, and other rats were passive, showing a significant freezing component in their behavior. These active and passive behavioral reactions to SD were negatively correlated. Testing animals 24 h after the SD procedure displayed behavioral and neurochemical changes, which were dependent upon a stress-coping strategy. Thus, the decreased immobility latency in the Porsolt test and cognitive augmentation in new object recognition assessment were found in rats with passive but not active coping. These behavioral adaptations in the passive coping animals were accompanied by enhanced electrically evoked DA release. No significant changes in DA release were observed in rats with active strategy during a confrontation with the aggressive opponent. Likewise, the dynamics of the DA decline under the depletion protocol were significantly altered in the passive but not active animals. The current study allowed us, for the first time, to link stress-evoked neurochemical consequences with specific behavioral changes in female rats.

From the methodological point of view, modeling SD stress in female rodents is quite challenging. Indeed, “social defeat” is a translational paradigm that is widely used to study stress-triggered behavioral and neurochemical alterations in male rodents [64,65,66,67]. This approach is prevalently based on the natural territorial and hierarchical aggression of animals. The exposure of a test subject to an aggressive opponent results in defensive and submissive behaviors that evidently point to stress due to a striking endocrine response [68]. In fact, most female rodents exhibit low levels of territorial and hierarchical aggression [69], which greatly complicates the ability to conduct SD experiments on female rats. Fortunately, tryptophan hydroxylase 2 (Tph2) knockout female rats with a genetically induced reduction in brain serotonin (5-HT) level demonstrated an escalated aggressive phenotype [29,30]. This specific feature of Tph2 knockout allowed us to create the condition where Wistar female rats could be reliably exposed to SD throughout the “intruder”–“resident” procedure.

Applying this procedure to female rats, we may observe a less homogenous response to stressful confrontations with aggressive opponents in male rats under identical circumstances [42]. As generally claimed, a female organism has different and more complex potential, compared to a male, to deal with stressful situations. Considering the critical role of unstable hormonal status due to a 4–5 days estrus cycle, it is easy to presume that stress-triggered changes in the female brain are more variable. In fact, there are multiple potential mechanisms by which the estrous cycle might influence stress- and anxiety-related behavior [37,70]. Existing research advocates a number of candidate downstream pathways stemming from alterations in ovarian hormones, estradiol and progesterone [71], which modulate the serotoninergic, oxytocinergic, GABAergic systems and the hypothalamus–pituitary–adrenal (HPA) axis [72,73,74,75]. In fact, all of these systems are interconnected with mesolimbic DA neurotransmission to some extent.

SD experiments indicated that some subjects expressed a significantly higher level of activity directed toward dealing with a stressful social event, while others were evidently passive, revealing a marked freezing pattern. Remarkably, these opposing behavioral responses to SD were negatively correlated. It is well documented that, similarly to humans, there are individuals among animals who preferentially perform actively or passively under a stress condition [18,76,77,78]. Therefore, the obvious difference in behavior during the stressful SD procedure allowed us to separate rats into actively and passively coping groups. The escalated freezing component coupled with decreased explorative activity observed in the behavioral response of passive rats may indicate a decrease in extracellular DA levels in the striatum. On the contrary, higher exploratory activity indicates increased extracellular DA concentrations in active subjects. In fact, the individual difference in the DA level that drives the behavior naturally occurs in rats and may shape the effects of drugs on the dopaminergic system [79,80] and a stress response [6,81,82,83,84]. Moreover, it was postulated that fluctuations in accumbal DA are responsible for the development of different (passive or active) coping strategies [6]. Therefore, an increased DA release in the nucleus accumbens is probably needed for efforts to escape or control stressful situations, while DA decline can promote a suppression of self-defensive behaviors. Behavioral responses observed in female rats with two opposite coping strategies are in line with this hypothesis. Importantly, the differences in the effect of social defeat, which are probably due to a specific individual variability of the tested phenotypes, may be causally linked with varying predispositions to stress. Thus, previous findings of studies in defeated male mice of an inbred strain indicated variable individual susceptibility to defeat [82]. In fact, susceptible mice exhibited a long-lasting upregulation in the firing rate of DA cells, while unsusceptible ones did not [82].

The main focus of the current study was on a short-term neuroadaptation that can be developed within 24 h following SD exposure. Previous data obtained in SD-exposed male rats demonstrated profound alterations in the evoked DA release in the nucleus accumbens [42]. In the present exploration, females with passive but not active coping strategies revealed similar consequences regarding DA transmission. Identically to the former study, increased DA efflux was found after the VTA stimulation at high frequencies (40, 50 and 60 Hz), whereas the lowest frequencies (5, 10, 20 and 30 Hz) did not induce significant changes compared to the control group. It should be pointed out that the group of male rats was two times smaller than the initial female group used in the present work. This probably did not allow us to distinguish contrasting behavioral responses to the SD environment from previous studies on males. However, we cannot exclude the possibility that male Sprague Dawley rats express more homogenous reactions to stressful confrontations with aggressive opponents than female Wistar due to their more uniform hormonal status and, perhaps, strain difference. Taken together with the common finding of increases in the DA measure, the behavior of male rats through SD exposure was more comparable to that expressed by female rats with passive coping in contrast to the active coping group. For example, both groups (males and passive females) demonstrated equal exploratory components and relatively similar freezing intensities. These data suggest a link between behavioral responses observed during SD and alterations in mesolimbic DA detected 24 h later. In fact, there was a negative correlation between active behavior and DA release in passively coping rats.

Rats who expressed active coping strategies and, perhaps, were less stressed than passive ones did not reveal adaptations in the observed DA release. Furthermore, in line with these neurochemical consequences, no significant behavioral changes were evident in active females. In contrast to behaviors expressed during SD exposure, the post-stress responses of passive coping rats suggest an augmentation in DA transmission. Surprisingly, this group demonstrated better performance in novel object recognition and Porsolt tests, as well as enhanced rearing in the maze. Indeed, one would expect that stress exposure should preferentially trigger abnormal rather than adaptive changes. Thus, our previous study in males indicated that the immobility time in the forced swim test was prolonged following the SD episode, suggesting a depression-like condition [42]. This inconsistency might be explained by the difference in the sex and strain. Furthermore, a more recent opinion considers the forced swim test as a test of coping strategy against an acute stressor [41]. Therefore, the data of the current study may shed light on the cause of variable results on learned helplessness by female rats.

There is convincing evidence that physiological responses to acute stress may improve short-term memory, fast learning and emotional status [85,86,87]. Notably, the data from human and animal studies showed that acute stress could result in cognitive disruptions in males, whereas it might enhance memory and induce hyperarousal in females [85,86,87,88]. Therefore, our results obtained in female rats are parallel to previous findings, which emphasize the positive consequences of acute stress on behavior.

Stress-induced alterations of DA release may result in consequent presynaptic adaptations, which are aimed at lowering an increased extracellular DA. Autoreceptor-mediated control of DA synthesis is often involved in such adaptations on a short-term scale [89]. To find whether the consequence of SD stress on DA efflux is capable of changing DA synthesis, we used the depletion protocol [42]. This procedure was established on the discovery that a certain amount of time is needed to recover DA release after the depletion induced by long electrical stimulation of the DA cell body region. We found that dynamics of the decay and recapture of accumbal DA under this condition are altered, indicating more powerful depletion and slower recovery in passive but not active rats. These results are consistent with the findings in humans that dopamine synthesis capacity correlated with the physiological response to an acute psychosocial stressor [21]. As was discussed earlier [21], the decline in synthesis can be explained by the activation of the inhibitory feedback mechanism through presynaptic autoreceptors. However, we cannot exclude other possible mechanisms. For example, acute corticosteroids, which regulate tyrosine hydroxylase activity, were associated with subsequent decreased striatal DA synthesis [90].

## 5. Conclusions

Our findings revealed that a single exposure of female rats to the SD paradigm might trigger a marked increase in accumbal DA efflux measured 24 h following the stressful event. In addition to this effect, we found decreased DA recovery that suggests a secondary decrease in the biosynthesis of the neurotransmitter. Remarkably, these neurochemical adaptations were dependent on the SD-coping strategy and associated with consequent behaviors. Therefore, the results support the hypotheses that individual coping styles and stress-induced plasticity within the VTA–nucleus accumbens circuitry may represent endophenotypes and biomarkers of susceptibility to stress [81,82].

## Figures and Tables

**Figure 1 biomolecules-14-01280-f001:**
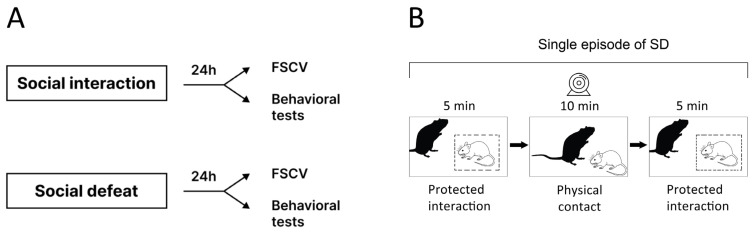
Schematic illustration of the experimental design: (**A**) Two groups of female Wistar rats were exposed to two different behavioral paradigms. In the first one, an intruder rat was reintroduced to its non-aggressive cagemate. These rats were housed together for at least 4 weeks. They interacted without any confrontation. In the latter paradigm, a subject was placed in the cage with a Tph2-KO resident. The intruder and aggressive resident were introduced to each other for the first time. All rats underwent either behavioral testing or in vivo fast-scan cyclic voltammetry (FSCV) measurements 24 h after SD. (**B**) Schematic illustration of a time course of the single SD procedure.

**Figure 2 biomolecules-14-01280-f002:**
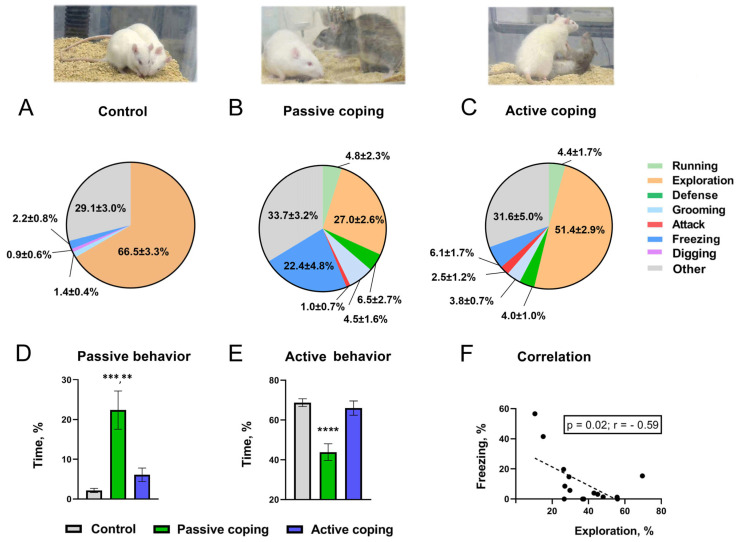
**Upper** panel: Behavioral patterns displayed by control rats during non-aggressive social interaction ((**A**), *n* = 20) and by intruder rats with passive ((**B**), *n* = 16) or active ((**C**), *n* =18) coping with the aggressiveness of Tph2-KO residents during the direct contact phase of single SD episode. **Lower** panel: Display of passive ((**D**), freezing) or active ((**E**), exploration + attacks + defense + running + grooming + digging) behavior by control and active or passive coping with SD stress rats. All data are mean ± SEM, expressed as a ratio (%) of total evaluation time, 10 min; levels of statistical significance: **—*p* < 0.01; ***—*p* < 0.001; ****—*p* < 0.0001. (**F**) Scatter plot of negative correlation between exploratory activity and freezing behavior of intruder rats under stress during the direct contact phase of SD.

**Figure 3 biomolecules-14-01280-f003:**
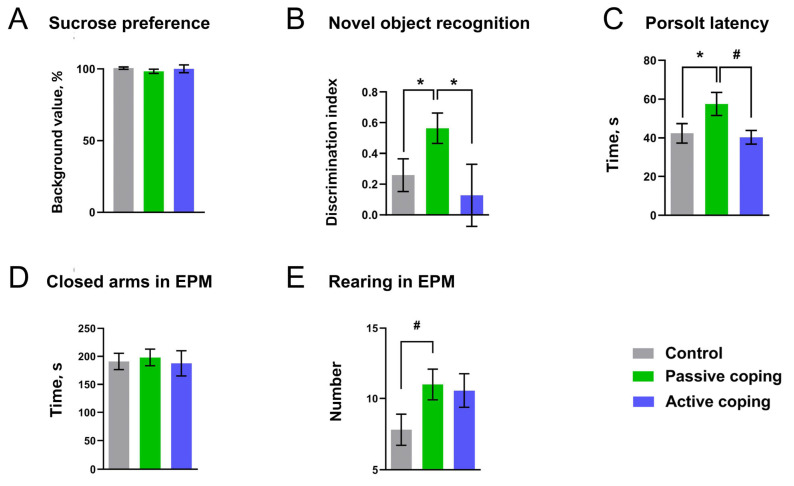
The behavior of rats displayed 24 h following non-aggressive social interactions (control) and SD episodes. Rats were tested in the standard battery of behavioral assessments, including (**A**) sucrose intake test, (**B**) novel object recognition, (**C**) Porsolt forced swim (time and latency of immobility) and (**D**,**E**) elevated plus maze tests. Color-coded columns were designated for the following experimental groups of rats: active coping (blue, *n* = 7), passive coping (green, *n* = 8) and control (grey, *n* = 15). All data expressed as mean ± SEM, * *p* ≤ 0.05; # *p* = 0.06.

**Figure 4 biomolecules-14-01280-f004:**
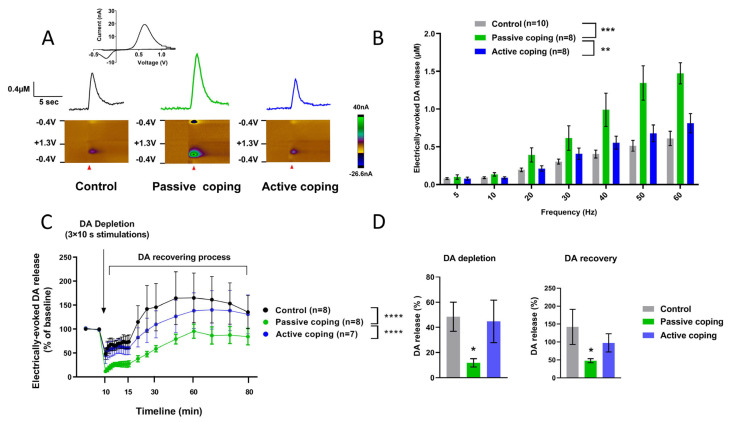
Alterations of electrically evoked DA response in rats with different coping observed 24 h after the single SD stress: (**A**) representative DA signals in response to the stimulation (60 Hz, 60 pulses, current 330 µA); (**B**) electrically evoked DA was released in a frequency-dependent manner in all tested groups, while the response was significantly higher in passively coping rats; (**C**) SD stress resulted in changes in DA efflux after the depletion (3 × 10 s, 60 Hz, 600 pulses, 330 µA) and during recovery processes; (**D**) changes (%) in DA depletion and recovery levels observed within 15 min following the VTA stimulation. Data are presented as mean ± SEM, repeated measures two-way ANOVA and Kruskal–Wallis test; *—*p* < 0.05; **—*p* < 0.01; ***—*p* ≤ 0.001; ****—*p* < 0.0001.

## Data Availability

The data presented in this study are available from the corresponding author upon reasonable request.

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
