# Peer review of "Accumbal Dopamine Responses Are Distinct between Female Rats with Active and Passive Coping Strategies"

_biomolecules, 2024, doi:10.3390/biom14101280_

Round 1

Reviewer 1 Report

Comments and Suggestions for Authors

In the study titled “Accumbal dopamine responses are distinct between female rats with active and passive coping strategies” Gainetdinov and colleagues investigated the short-term effects of a single social defeat (SD) on dopamine (DA) dynamics in the nucleus accumbens and related behaviors in female Wistar rats. It is a continuation of previous research conducted by this group on male rats, where they demonstrated that stress induced by a single SD is capable of generating behavioral changes and dopamine (DA) alterations in the nucleus accumbens. The authors observed two patterns of stress-coping abilities: active and passive. Active copers showed more activity during stress, while passive copers exhibited increased freezing behavior. Twenty-four hours after SD, passive rats displayed reduced immobility latency, improved cognitive performance, and increased mesolimbic DA release, whereas active rats showed no significant changes. This suggests that females with passive coping strategies are more susceptible to stress-induced neurochemical and behavioral changes. I believe this study is important for researchers as it highlights differential stress responses in females, emphasizing the need for gender-specific approaches to the diagnosis and treatment of stress-related disorders.

There are a few issues that should be addressed prior to publication in Biomolecules

Abstract

The Abstract section clearly describes the main findings of the study. The terminology regarding active/passive coping strategies should be refined for consistency with the context. For instance, in row 19, the phrase "rats with an active strategy" could be reconsidered to reflect more precise language, such as "rats exhibiting an active pattern of responses." The last sentence/conclusion should be revised to deliver clear message of the study concussions.

Introduction

The introduction is well-structured. I suggest also addressing the information on stress-coping abilities from established mouse models of inherited stress vulnerability, focusing on gender-dependent metabolic and behavioral parameters, as well as lifespan .

Methods.

The study design is well-structured, and the assessment of the estrus cycle is a great asset to the study's fidelity. Please also state (in row 88) whether the intruder and resident were of the same age/weight.

Regarding rows 95-96, how can you ensure that the "normal" Wistar female intruder does not exhibit innate exaggerated aggressive behavior? Please elaborate.

In Figure 1A, the difference between the "social interactions" and "social defeat" groups should be further clarified, at least in the figure legends

Results.

Row 207-209 – the sentence should be revised

Figure 4D should be further clarified, as it is difficult to understand the specific timeframe during which dopamine release was measured.

Discussion

The discussion clearly addresses the major findings of the study.

In rows 308-309, while I generally agree with this statement, there are situations where a dominant individual, when exposed to aggressive individual, may experience different outcomes. It could even result in the subordination of the 'aggressive' resident.

Author Response

Dear Editor,

We appreciate your consideration of our manuscript and the opportunity to further improve it. All concerns have been completely addressed; therefore, the quality of the manuscript is greatly enhanced. We hope the manuscript is now suitable for acceptance in Biomolecules. A summary of our responses to the referees’ comments follows:

Reviewer 1

Question 1:

The Abstract section clearly describes the main findings of the study. The terminology regarding active/passive coping strategies should be refined for consistency with the context. For instance, in row 19, the phrase "rats with an active strategy" could be reconsidered to reflect more precise language, such as "rats exhibiting an active pattern of responses." The last sentence/conclusion should be revised to deliver clear message of the study concussions.

Response 1:

As the Reviewer requested, we provide more precise language for the terminology regarding coping strategies (line 19-20). We added the new sentences in order to deliver clear message of the study concussions (line 19-20).

Question 2:

The introduction is well-structured. I suggest also addressing the information on stress-coping abilities from established mouse models of inherited stress vulnerability, focusing on gender-dependent metabolic and behavioral parameters, as well as lifespan (for examples see PMID: 33601678, PMID: 31707362).

Response 2:

The information on stress-coping abilities from established mouse models of inherited stress vulnerability, focusing on gender-dependent metabolic and behavioral parameters, as well as lifespan was provided (lines 41-43)

Question 3:

A - The study design is well-structured, and the assessment of the estrus cycle is a great asset to the study's fidelity. Please also state (in row 88) whether the intruder and resident were of the same age/weight.

B - Regarding rows 95-96, how can you ensure that the "normal" Wistar female intruder does not exhibit innate exaggerated aggressive behavior? Please elaborate.

C - In Figure 1A, the difference between the "social interactions" and "social defeat" groups should be further clarified, at least in the figure legends

Response 3:

A – We confirmed that Intruder and resident female rats were the same weigh and age (line 79-80).

B - Indeed, the most female rodents exhibit low levels of territorial aggression (this is well documented phenomena). Female intruder rats express such aggression even at lesser extent, since it is not their territory. We elaborate it in previous version of the Discussion (Line 328-330).

C – We further clarified the difference between the "social interactions" and "social defeat" groups in the figure legends (Figure 1).

Question 4:

A - Row 207-209 – the sentence should be revised

B - Figure 4D should be further clarified, as it is difficult to understand the specific timeframe during which dopamine release was measured.

Response 4:

A - The sentence was revised (line 221-223).

B – The figure was clarified by adding the information on time frame of dopamine release recordings (4D figure legend).

Question 5:

The discussion clearly addresses the major findings of the study.

 In rows 308-309, while I generally agree with this statement, there are situations where a dominant individual, when exposed to aggressive individual, may experience different outcomes. It could even result in the subordination of the 'aggressive' resident.

Response 5:

We are pleased to see that the Reviewer accepted our statement. We agree with the Reviewer that situations with different outcomes could have a place too.

We do appreciate professional and constructive assessment of our report by reviewers that undoubtedly helped us to improve the manuscript.

Reviewer 2 Report

Comments and Suggestions for Authors

This study presents an effective protocol for studying social defeat in female rats. Indeed, as the authors explain, animal models testing the psychobiological effects of social stress in females are needed. The methods used for this study are sound, and the results are interesting and stimulating. My comments aim to improve their presentation to the scientific community.

The study does not directly compare the effects of the experimental protocol in male and female rats. This should be stated in the ‘Limitations’ section of the paper. Nonetheless, this research group recently tested this same defeat protocol in male Sprague Dawley rats. Comparing the results obtained in this previous study with those reported in the submitted paper indicates that male rats show extreme passive coping during defeat and high helplessness levels as measured by FST and enhanced electrically-stimulated DA release in the accumbens 24 hrs later, whereas a population of females from the same rat strain defeated at the same age (based on reported weight) does not.

This information should be given in the abstract and the introduction because this is a sex-dependent difference.

The discussion should include the following considerations: 1) The study tested whether the sex-dependent differences in the effects of defeat were due to a sex-specific individual variability of the tested phenotypes in female rats. Indeed, findings of studies in defeated male mice of an inbred strain (therefore sharing the same genotype) indicate variable individual susceptibility to defeat (DOI: 10.1016/j.cell.2007.09.018). 2) The results obtained support this hypothesis as well as the hypotheses that individual coping styles and stress-induced plasticity within the mesolimbic DA system represent endophenotypes and possible biomarkers of individual susceptibility to the adverse effects of stress (DOI: 10.1016/j.cell.2007.09.018; DOI: 10.3389/fnbeh.2021.785739).

3) Results obtained by the study shed light on the cause of variable results on the development of learned helplessness by female rats (DOI: 10.3758/s13415-024-01171-2, for a recent review). Indeed, learned helplessness describes the bias toward expressing a passive coping strategy in the presence of a novel escapable stressor following a previous experience of inescapable stress that is consistently reported in male rats. The observation that female active copers express reduced helplessness in FST 24 hrs after defeat supports the view that the development of learned helplessness by female rats depends on the percentage of active copers included in the tested population.

Comments on the Quality of English Language

The quality of the English Language used is sufficient. However, it can be improved to help the reader to appreciate the paper fully

Author Response

Dear Editor,

We appreciate your consideration of our manuscript and the opportunity to further improve it. All concerns have been completely addressed; therefore, the quality of the manuscript is greatly enhanced. We hope the manuscript is now suitable for acceptance in Biomolecules. A summary of our responses to the referees’ comments follows:

Reviewer 2

Question 1:

The study does not directly compare the effects of the experimental protocol in male and female rats. This should be stated in the ‘Limitations’ section of the paper. Nonetheless, this research group recently tested this same defeat protocol in male Sprague Dawley rats.

 Comparing the results obtained in this previous study with those reported in the submitted paper indicates that male rats show extreme passive coping during defeat and high helplessness levels as measured by FST and enhanced electrically-stimulated DA release in the accumbens 24 hrs later, whereas a population of females from the same rat strain defeated at the same age (based on reported weight) does not.

This information should be given in the abstract and the introduction because this is a sex-dependent difference.

Response 1:

As the Reviewer requested, we stated in the Limitation section that the study does not directly compare the effects of the experimental protocol in male and female rats (Lines 199-206).

Question 2:

The discussion should include the following considerations:

  1. A) The study tested whether the sex-dependent differences in the effects of defeat were due to a sex-specific individual variability of the tested phenotypes in female rats. Indeed, findings of studies in defeated male mice of an inbred strain (therefore sharing the same genotype) indicate variable individual susceptibility to defeat (DOI: 10.1016/j.cell.2007.09.018).

  1. B) The results obtained support this hypothesis as well as the hypotheses that individual coping styles and stress-induced plasticity within the mesolimbic DA system represent endophenotypes and possible biomarkers of individual susceptibility to the adverse effects of stress (DOI: 10.1016/j.cell.2007.09.018; DOI: 10.3389/fnbeh.2021.785739).
  2. C) Results obtained by the study shed light on the cause of variable results on the development of learned helplessness by female rats (DOI: 10.3758/s13415-024-01171-2, for a recent review). Indeed, learned helplessness describes the bias toward expressing a passive coping strategy in the presence of a novel escapable stressor following a previous experience of inescapable stress that is consistently reported in male rats. The observation that female active copers express reduced helplessness in FST 24 hrs after defeat supports the view that the development of learned helplessness by female rats depends on the percentage of active copers included in the tested population.

Response 2:

Based on the Reviewer comments, we included the following considerations:

           A – The indication of variable individual susceptibility to defeat (Lines 365-370).

  B - Individual coping styles and stress-induced plasticity within the mesolimbic DA               system represent endophenotypes and possible biomarkers of individual susceptibility (Lines 433-435).

  C - Results obtained by the study shed light on the cause of variable results on the development of learned helplessness by female rats (Line 402-405).

We do appreciate professional and constructive assessment of our report by reviewers that undoubtedly helped us to improve the manuscript.